# Immune Responses to Sequential Binocular Transplantation of Allogeneic Retinal Progenitor Cells to the Vitreous Cavity in Mice

**DOI:** 10.3390/ijms24076205

**Published:** 2023-03-25

**Authors:** Lu Chen, Jing Yang, Henry Klassen

**Affiliations:** 1Department of Ophthalmology, University of California, 845 Health Sciences Road, Irvine, CA 92697, USA; 2Enochian BioSciences, Los Angeles, CA 90036, USA; 3jCyte, Inc., Newport Beach, CA 92660, USA

**Keywords:** stem cells, immune tolerance, immune privilege, graft rejection, intravitreal injection

## Abstract

Intravitreal transplantation of allogeneic human retinal progenitor cells (hRPCs) holds promise as a treatment for blinding retinal degenerations. Prior work has shown that neural progenitors are well-tolerated as allografts following single injections; however, sequential delivery of allogeneic cells raises the potential risk of host sensitization with subsequent immune rejection of grafts. The current study was designed to assess whether an immune response would be induced by repeated intravitreal transplants of allogeneic RPCs utilizing the mouse animal model. We injected murine retinal progenitor cells (gmRPCs), originally derived from donors with a C57BL/6 genetic background, into BALB/c recipient mice in order to provide safety data as to what might be expected following repeated treatment of patients with allogeneic human cell product. Immune responses to gmRPCs were mild, consisting of T cells, B cells, neutrophils, and natural killer cells, with macrophages clearly the predominating. Animals treated with repeat doses of gmRPCs did not show evidence of sensitization, nor was there immune-mediated destruction of the grafts. Despite the absence of immunosuppressive treatments, allogeneic gmRPC grafts survived following repeat dosing, thus providing support for the preliminary observation that repeated injection of allogeneic RPCs to the vitreous cavity is tolerated in patients with retinitis pigmentosa.

## 1. Introduction

Retinitis pigmentosa (RP) encompasses a class of genetic rod-cone degeneration and results in progressive visual impairment and eventual blindness. For the vast majority of cases, there are no proven therapeutic measures available to preserve or restore vision; however, one strategy for addressing this unmet medical need is stem cell transplantation. Our research group has explored intravitreal injection of cultured human retinal progenitor cells (hRPCs) as a way to intervene in RP with the therapeutic goal of stabilizing progression or perhaps reversing the course of the disease. hRPCs can be derived from immature retinas via tissue donation or, more recently, from pluripotent cell lines. Studies in animal models have shown that these cells are capable of rescuing photoreceptors from degeneration following injection into the eye and are also capable of differentiating into rod photoreceptors in the host eye. There is some evidence from animal studies that injected hRPCs can become functional, integrated photoreceptors and thereby potentially stabilize the retina by directly replacing dying cells. Thus, injected hRPCs might treat RP in both a neurotrophic fashion as well as via a cell replacement mechanism. Either way, this cell-based approach offers a rational strategy for treatment of patients afflicted with RP and, potentially, other retinal diseases.

Another favorable attribute of RPCs, and perhaps neural progenitors in general, is the relative tolerance afforded these cells following transplantation as allografts, particularly when delivered to a location exhibiting immune privilege such as the retina [1,2]. That said, recent reviews of clinical trials conducted in the field of regenerative medicine continue to report widespread and significant challenges related to inflammation and immune rejection [3,4], including for interventions targeting the eye such as retinal gene therapies [5], as well as some but not all cell therapies [6]. Although a local cell type, the retinal pigment epithelial (RPE) cell is not exempt from rejection issues [7,8,9]; therefore, immune suppression of recipients is currently standard practice [10,11]. Human RPCs, on the other hand, appear to be remarkably well-tolerated as allografts [6,12,13,14,15,16], although this does not extend to their use as xenografts, where immune suppression is required [17].

The basis for sustained survival of allogeneic RPCs in the eye is likely to involve a number of factors related to the cells used and the recipient site [1,8,18,19]. In previous work using flow cytometry, we have shown that human neural progenitors, both brain-derived and retina-derived, express class I, but not class II, major histocompatibility (MHC) antigens [20,21]. The classical mechanism of graft rejection involves the nonspecific recognition of foreign MHC class II molecules by CD4+ host lymphocytes [20,21,22,23,24,25]. In that way, an absence of class II molecules might allow grafted progenitor cells to evade immune rejection mediated via this mechanism. However, the apparent “immune privileged” status of intravitreally injected hRPCs is not necessarily absolute. While murine central nervous system (CNS) progenitors do not express detectable class I or class II MHC molecules at baseline and exhibit apparent immune privilege as allografts [1,2,20], these cells can undergo immune rejection under certain circumstances. Studies have shown that MHC antigens can be induced through the stimulation of CNS progenitors with interferon-gamma (IFNγ) [1,26]. CNS progenitors can be rejected following sensitization of a previously grafted host. Therefore, CNS progenitor cells (including RPCs) do express alloantigens that are detected by the host immune system. Together, this implies that the immune privileged status of grafted progenitor cells is provisional and subject to modulation, e.g., by cytokines present in the local microenvironment, and which can undergo change during the course of degenerations such as RP. For all these reasons, the immune response to multiple intravitreal RPC injections is difficult to predict and must be specifically examined.

If hRPC treatment proves to be clinically successful in RP, there will be strong incentives for treatment of both eyes in this bilateral blinding disease. For reasons of safety at the very least, binocular treatments would typically be performed sequentially, with a significant lag time between the two injections. Of note, this sequential delivery of allogeneic cells might result in host sensitization in response to the first injection, potentially triggering immune rejection in response to the second injection. This could result in the loss of grafts in both eyes. Prior reports examining single subretinal dosing with hRPCs in RP [27] or redosing with human neural progenitors in animals [28,29] incorporated immune suppression. Therefore, the following study was designed as a translational investigation of murine RPCs as sequential allografts in an immune competent mouse strain with a disparate genetic background. While not including human RPC product, this work was performed as part of IND-enabling studies in order to provide animal data addressing safety, i.e., whether immune suppression would be necessary for patients receiving repeated RPC allograft treatments in FDA-registered trials.

## 2. Results

To study the immunological consequences of bilateral RPC injection in allogeneic mice, gmRPCs (C57BL/6 genetic background) were injected intravitreally into one eye of BALB/c recipients and a second dose of RPCs was injected into the other eye 2 weeks later. The 2-week time interval was chosen to be sufficient to allow the first graft to induce both innate and adaptive immune responses. None of the experimental animals received any immune suppressive drug treatments so that the natural physiological immune responses could be observed.

### 2.1. Clinical Observation and Ophthalmic Examination Revealed No Abnormalities

The weights of experimental animals were measured at the time of each gmRPC injection. All experimental animals showed weight gain between the first and second injection, consistent with overall good health. Animals that were terminated at day 14 and day 28 ± 1 after the second gmRPC injection were examined through the surgical microscope. Both the anterior and posterior segments of the majority of the examined eyes were judged to be “within normal limits” (wnl). There were no signs of inflammation or other visible abnormalities within untreated, sham-treated, and gmRPC-treated groups, indicating that gmRPC intravitreal injection did not illicit inflammatory responses or physical tissue damage detectable via this method. Cell clusters comprising the graft were visible in the vitreous at both day 14 and day 28 ± 1 following the second gmRPC injection (Figure 1). Repeated injection of gmRPCs did not induce noticeable changes in the size of the cell clusters, consistent with sustained donor cell survival. Gross pathology was also evaluated at the terminal end point and no enlarged eyeballs (indicative of tumor formation) or other abnormalities were noted.

### 2.2. Immunofluorescent Labeling Showed Immune Cells in the Vitreous

Immunofluorescent labeling for five major immune cell types (macrophages, neutrophils, natural killer cells, T cells, and B cells) was performed on the ocular cryosections and revealed positive immune cell infiltration in the eye following gmRPC injection (Figure 2A–F). T cell (CD3), B cell (CD45R), neutrophil (Ly-6G), and natural killer cells (CD49b) exhibited a limited presence in eyes injected with gmRPCs, whereas activated macrophages (Iba-1) were the main immune cells responding to the intravitreal grafts. Comparable tissue from untreated and sham-treated animals did not show immune cells in ocular cryosections, suggesting that the immune cell infiltration seen was a response to the gmRPC grafts. Consistent with this observation, the immune cells were located in the area surrounding the gmRPC grafts or within the gmRPC clusters themselves.

The peak numbers of each immune cell in the eye varied by cell type (Figure 3A–E). In contrast, two-way ANOVA revealed no differences in immune cell infiltration when comparing animals that received single gmRPC injections (sham/cells) to those that received bilateral injections (cells/cells). This was the case for infiltration of macrophages (anti-Iba-1 staining) (Figure 3A), neutrophils (anti-Ly-6G staining) (Figure 3B), natural killer cells (CD49b staining) (Figure 3C), T cells (CD3 staining) (Figure 3D), and B cells (CD45R staining) (Figure 3E).

### 2.3. ELISPOT Did Not Show Antigen Recall Responses

ELISPOT is a way to assay antigen specific IFNγ producing T cell activation. Phorbol 12-myristate 13-acetate (PMA) treatment was a positive control mimicking the second messenger, DAG, to activate the T cell receptor pathway and, in turn, causing T cell activation. In Figure 4A–D, PMA treated groups showed much higher IFNγ spot counts compared to control groups with responder cells alone (i.e., lymphocytes from CLNs, splenocytes from spleens). When the responder cells were co-cultured with C57BL/6 splenocytes (B6 SPL), the day 14 samples were comparable to the responder cells alone groups, with only slightly higher IFNγ spot counts, indicating a lack of T cell activation (Figure 4A,B).

When testing allogenic donor cells using this assay, the responder cells from the animals intraperitoneally injected with gmRPCs (IP group) did not have antigen recall responses since the samples did not show more T cell activation by IFNγ spot counts compared to the sham animals never exposed to gmRPCs (sham/sham). Neither did the animals that were intravitreally injected with the gmRPCs, either once (sham/cell) or repeatedly (cell/cell) show any antigen recall responses. In one case, responses to repeat gmRPC dosing appeared more muted (Figure 4B). Together, these results indicated that gmRPCs exhibit very low antigenicity.

Note that C57BL/6 splenocytes were matched with the gmRPCs’ genetic background. Unexpectedly, the alternative stimulator cells, gmRPCs, increased the IFNγ spot count in all the experimental groups compared to negative control groups (Figure 4A–D). A potential explanation might be the gmRPCs causing high background in this assay. Importantly, there was no difference between the untreated, sham-treated, single gmRPC injection, and repeated gmRPC injection groups, which again indicated an absence of antigen recall responses.

## 3. Discussion

There has been considerable interest in stem cell technology as a means of treating retinal degenerative diseases and our group has been exploring the use of allogeneic hRPCs in RP. To gather data addressing potential immune consequences when more than one dose of allogeneic human RPC product is administered via intravitreal injection, we carried out the present animal study using gmRPCs from a C57BL/6 background as grafts and BALB/c mice as hosts. These disparate genetic backgrounds were used to model sequential treatment with allogeneic hRPCs and provide preliminary safety data prior to testing in humans.

To fully evaluate responses, host animals did not receive any immuosuppressive treatment. We compared the immune responses generated by a single gmRPCs injection to a sequential bilateral dosing according to the injection schedule described. Ophthalmic examinations did not reveal inflammation or any other notable differences between the groups of animals that received 1 dose of gmRPCs or 2 doses of gmRPCs, suggesting that either recipients were not sensitized by the initial gmRPC treatments, or the sensitization was subtle. Immunolabeling results revealed that all five immune cell types examined, namely T cells (CD3), B cells (CD45R), macrophages (Iba-1), neutrophils (Ly-6G), and natural killer cells (CD49b), had infiltrated into the eye and targeted the RPC grafts post-transplantation. Nevertheless, this infiltration was relatively moderate and, although other cell types were detectable, it was predominantly composed of Iba-1 positive macrophages. A very limited numbers of T cells, B cells, neutrophils, and natural killer cells was present. Interestingly, the survival demonstrated by single and repeat RPC grafts was equivalent, without loss of grafts in either case, despite the presence of macrophages. Overall, this suggests that both innate and adaptive immune responses to genetically disparate RPCs were moderate, even though immune suppression was not employed.

Furthermore, ELISPOT assay results showed no antigen recall responses in gmRPC treated groups, consistent with the contention that T cells do not play an important role in rejection of gmRPC grafts from C57BL/6 mice. Again, macrophage infiltration of gmRPC grafts was observed, although this was not associated with significant graft destruction of the type associated with classical immune rejection [22,23,24].

Together, these observations reveal a situation that clearly differs from the widely recognized T and B cell mediated allogeneic transplant rejection mechanisms seen in other tissues. Multiple factors could contribute to this situation. On the one hand, there is the notion of the eye as an “immune privileged” site, even though the extent to which the vitreous shares such characteristics with the cornea and retina is less studied and perhaps more contentious. On the other hand, the RPCs themselves appear to exhibit decreased immunogenicity as allografts, compared to more frequently studied cell types. As we have shown previously, murine RPCs lack baseline expression of MHC class I and class II antigens, although these are inducible via cytokine stimulation. This absence of MHC expression might contribute to the survival of these cells following transplantation.

Nevertheless, the presence of immune cells in the grafts supports the concept that graft tolerance in this setting is not passive and results from an active immunoregulatory phenomenon. Here, it should be emphasized that the same sort of immune cell infiltration was already present following single injections and was not exacerbated by repeat dosing, thus underscoring our conclusion that immune cells infiltration of the grafts, particularly by macrophages, was not the result of prior host sensitization. Within the parameters of this experiment, repeat dosing did not appear to diminish overall graft viability, which was consistent with the possibility of sustained efficacy in therapeutic contexts. Alternate parameters, such as different inter-injection intervals, might either increase or lessen immune cell activity to an extent that would be relevant to clinical application. Whether the cellular constituents of infiltrates, including macrophage predominance, would also apply to humans is unclear. Furthermore, the survival of these allografts might not be dependent on a normal retina with a healthy blood-retinal barrier. The BALB/c recipients used here are albinos and susceptible to light damage [30]. Moreover, sustained survival has been seen repeatedly in previous work with various retinal degeneration models (e.g., [31]).

Compared to graft survival, the viability of cells within the grafts presents a more complex situation. There is a known loss of CNS progenitor cells that occurs rapidly post-transplantation, likely due to multiple factors potentially including abrupt growth factor withdrawal. Once settled within the eye, the dissociated cells coalesce into sphere-like aggregates, the centers of which appear to provide a suboptimal microenvironment for growth, perhaps due to the lack of vascularization and challenges to nutrient diffusion. The apparent tropism of macrophages for the RPC grafts could be a nonspecific response to the presence of these nonviable cells in the interior of the grafts, with the macrophages functioning in their role as scavengers of cellular debris. Alternatively, fully viable RPCs might actively attract the macrophages, e.g., via expression of chemo-attractive cytokines, whereupon the host cells would secondarily encounter the debris from non-viable donor cells. Importantly, the degree of macrophage infiltration seen in these particular settings does not appear to have negative consequences for the graft or the host retina, in contrast to responses to cellular allografts in non-immune privileged sites [4].

Finally, it is worth mentioning that the results reported here, although restricted to a murine model, may have implications for work in humans. In contrast to the mouse, we know that cultured human RPCs do express robust levels of MHC class I antigens at baseline; therefore, the comparison is admittedly tentative. Nevertheless, it is interesting to note that clinical testing of subretinal hRPC grafts by a group in China [13], as well as intravitreal hRPC transplantation by our group, has subsequently shown sustained survival of the allografts in non-immunosuppressed patients with RP (JC-01, [32], unpublished data). This was also the case following sequential injection of both eyes (JC-01 Extension, unpublished data), analogous to the work presented here. In addition, graft survival was seen in a follow-on study on JC-02 [33], in which sequential repeat doses were administered to the same eye [34]. Taken together, these findings indicate that immune suppression is not always necessary in the setting of neural progenitor transplantation, particularly to the eye, although the limits of this phenomenon and underlying regulatory mechanisms remain to be elucidated.

## 4. Materials and Methods

### 4.1. Cell Culture

Previously characterized gmRPCs [31] were chosen as donor cells for this allogeneic study. Originally, gmRPCs were isolated from GFP-transgenic C57BL/6 mice genetically modified to express enhanced green fluorescent protein (GFP). The expression of GFP protein in the gmRPCs was of interest in that it allows visualization of grafts following injection without the need for additional labeling. For the current study, gmRPCs were cultured in Advanced DMEM/F12 supplemented with N-2 Supplement (1:100, Life Technology, Carlsbad, CA, USA), Glutamax-1 (1:100, Life Technology, Carlsbad, CA, USA), and EGF (20 ng/mL, recombinant, Human, Life Technologies, Carlsbad, CA, USA) and incubated at 37 °C and 5% CO_2_. The cells were maintained in mixed suspension/loosely attached colonies in uncoated tissue culture flasks. The cells were passaged via trypsinization with TrypLE Select CTS (Life Technologies, Carlsbad, CA, USA) diluted 1:5 in PBS for one minute and neutralized by adding 10 times the volume of trypsin added. The cell-trypsin mix was centrifuged at 140 g for 2 min at room temperature, the supernatant removed, and the cell pellet resuspended in fresh media before seeding to tissue culture flasks at the desired concentration. Cell concentration and viability was determined via trypan blue staining; counts were performed by Countess (Life Technologies, Carlsbad, CA, USA) and hemocytometer.

For intravitreal injections, the cell pellet was resuspended at 50K cells/μL in BSS PLUS^®^. Cell concentration and viability were evaluated before and after injections. The dose injected was 50,000 cells per eye per injection. The dose and vehicle were selected based on our previous studies.

### 4.2. Experimental Animals

The purpose of the current study was to investigate the potential allogeneic immune responses induced via repeat intravitreal injection of allogeneic RPCs. The recipient animals were BALB/c mice, which differ genetically from the C57BL/6 mice that the gmRPCs were isolated from.

The use of at least three animals per treated experimental group for each time point was deemed to be needed to allow for evaluation of statistical significance between groups with respect to outcome variables and potential animal losses during the experimental period. Furthermore, given the many additional factors potentially affecting the experimental outcome, such as unsuccessful cell injection procedure or potential attrition as result of animals fighting, injection procedures were performed on five animals per group to further ensure meeting the minimum numbers required for statistical analysis.

Table 1 shows the four groups of animals prepared for each of the four time points (4, 7, 14 and 28 days) at which evaluations of histopathology, including immunofluorescence, would be performed. Two additional groups of animals injected with 10^6 gmRPCs intraperitoneally were prepared as positive controls for each of the two ELISPOT evaluation time points (14 and 28 days). Table 2 summarizes the evaluations scheduled for each group.

Mice aged 5 weeks old BALB/c were purchased from Charles River Laboratories. Sexes of the animals were mixed and matched to ensure similar numbers of male and female animals were tested for each condition. All animals were kept in an animal facility with barrier-housed cages for a week before any procedures were performed on them.

### 4.3. Intravitreal Injection

The gmRPCs were administered via intravitreal injection. Once anesthetized, Mydriacyl (1% Tropicamide ophthalmic solution) and Phenylephrine (2.5% ophthalmic solution) were applied to both eyes of the animal to be injected. When adequate mydriasis had been achieved, the animal was restrained manually and the animal’s head rotated to align the ocular axis of the eye to be injected with the optics of the surgical microscope so as to visualize the ocular posterior segment (i.e., vitreous and retina). The eye was gently proptosed manually via pressure on the eyelids and a 31G needle on an insulin syringe was used to gently pierce the sclera adjacent to the limbus in the inferior nasal quadrant. A polished glass micropipette tip containing donor cells (or vehicle control) was advanced through the incision into the vitreous cavity under direct visualization. Care was taken not to disrupt the integrity of the lens or posterior segment structures. The cells or vehicle alone were injected in a volume of 1 microliter. After several moments of pause to allow equilibration of intraocular pressure, the pipette tip was gradually removed from the eye, all under direct visualization. Any intraocular bleeding was noted, along with location.

Following injection, the animal was placed in a clean cage lined with a fresh disposable pad oriented with absorbent-side-up/plastic-side-down, to recover. Recovery was facilitated using a heat pad and was verified by the ability of the animal to ambulate. After waking, the animal was transferred to the post-operative cage with ad libitum access to water. Animals were visually monitored on a daily basis post-operatively. Minimal or no signs related to intraocular injection were observed.

After the procedure, the animals were observed for rubbing of the operated eye, ruffled fur, or sustained lethargy, all of which were grounds for immediate euthanasia. There were two animals terminated as a result of wounds sustained in fighting. All other animals were terminated at the planned end points. Euthanasia was performed via CO_2_ inhalation.

### 4.4. Ophthalmic Examination

A Leica Ultimate Red Reflex Surgical Microscope was used for ophthalmic examination and photography of experimental mice. Sedation was performed with a Ketamine Hydrochloride/Xylazine Hydrochloride mixture (50–100 mg/mL Ketamine, 5–10 mg/mL Xylazine) administered by intraperitoneal injection. After sedation, topical mydriatics (Tropicamide, Phenylephrine) were applied to the eye(s) to be imaged in 5 min intervals until adequate mydriasis was achieved, as determined via pupil diameter (>2.5 mm) and response to light stimulation. As compensation for the loss of blinking, hypromellose (Gonak) solution was applied topically as needed to both eyes to prevent corneal desiccation. For the imaging procedure, anesthetized animals were placed on a heating pad and positioned in sternal recumbency. At the conclusion of the procedure, partial reversal of anesthesia was achieved with administration of Atipamezole (0.1–1 mg/kg) by intraperitoneal injection.

### 4.5. Histopathology

Immune responses in animals receiving two sequential grafts (one in each eye) were evaluated via histopathology of both eyes at day 4, day 7, day 14, and day 28 ± 1 following injection of the second eye. The mice that were treated according to Table 1 were terminated based on the stated schedule in Table 2. Animals were euthanized using carbon dioxide inhalation. Cardiac perfusions were performed on the experimental animals with 2% paraformaldehyde (PFA) in phosphate-buffered saline (PBS). Then, all the eyeballs were collected and allowed to soak in 2% PFA/PBS for 48 h at 4 °C. The fixed eyeballs were processed through a sucrose gradient (10% sucrose in PBS for 1 h at room temperature, 20% sucrose in PBS for 1 h at room temperature, and 30% sucrose in PBS at 4 °C overnight) before being embedded in OCT media for cryosectioning. Cryosections (10 μm) of the eyes were stained with Harrison hematoxylin and eosin (H&E) to visualize retinal microanatomy and locate the injected donor cells.

For the gmRPC treated groups, cryosections were evaluated using immunofluorescence for GFP+ (donor) cells. Identification of the specific immune cell types present in the eyes was carried out through labeling with specific primary antibodies for the following: CD3 (T lymphocyte marker) [35], CD45R (B lymphocyte marker) [36], Iba-1 (activated macrophage and microglia marker) [37], Ly-6G (neutrophil marker) [38], and CD49b (natural killer cell marker) [39]. Cryosections were washed in PBS three times at room temperature and blocked with 0.03% Triton X-100 and 10% normal goat serum in PBS (NGS; Jackson ImmunoResearch, West Grove, PA, USA) for 1 h at room temperature. Rat anti-mouse CD3 (1:100 dilution, BD Biosciences, San Jose, CA, USA), rat anti-mouse CD45R (1:100, BD Biosciences, San Jose, CA, USA), rabbit anti-mouse Iba-1 (1:400 dilution, Wako Chemicals, Richmond, VA, USA), rat anti-mouse Ly-6G (1:100 dilution, BD Biosciences, San Jose, CA, USA), and rat anti-mouse CD49b (1:100 dilution, BD Biosciences, San Jose, CA, USA) were applied to samples and incubated overnight at 4 degrees. Samples were washed again in PBS 3 times at room temperature. Alexa-Fluor-conjugated secondary antibodies (goat anti-rat Alexa-Flour-568 and goat anti-rabbit Alexa-Flour-568, Life Technologies, Carlsbad, CA, USA) were applied to samples for 1 h at room temperature. All samples were then washed in PBS for another three times at room temperature. Coverslips were mounted using DAPI Fluoromount-G (Southern Biotech, Birmingham, AL, USA) and the slides were left to dry overnight at room temperature.

### 4.6. ELISPOT and MLR Assays

RPC-specific memory T cells were monitored at 2 and 4 weeks after the first graft placement via an enzyme linked immunosorbent spot (ELISPOT) assay. The experimental groups were setup as shown in Table 1 and Table 2 for each time point. ELISPOT assay captures secreted proteins on a specific antibody-coated microplate and can be used to determine memory T cell activation by detecting T cell IFNγ secretion. The experimental readout is the number of the IFNγ spots on the microplate. The number of IFNγ spots present is indicative of the number of T cells being activated.

The frequency of alloreactive T cells was assessed by performing a 48 h mixed leukocyte reaction (MLR) on 96-well Multiscreen-IP plates (Millipore, Burlington, MA, USA). Responder cells were lymphocytes/splenocytes containing T cells purified from cervical lymph nodes (CLNs) and spleens of the experimental animals, and stimulator cells were the gmRPCs originally derived from C57BL/6 mice. The ELISPOT assays were carried out based on standard protocol. 96-well Multiscreen-IP plates were pre-wetted with 15 μL of 35% ethanol per well under sterile conditions. The plates were washed with sterile PBS three times before 100 μL of the IFNγ capture antibody (clone AN-18, eBioscience, San Diego, CA, USA) diluted in PBS (2 μg/mL) was applied to the plates. The plates were incubated overnight at 4 degrees before the MLR assays were carried on the plates. The capture antibody coated plates were washed three times with PBS and then blocked with RPMI1640 for 2 h at 37 °C.

Each of the assay plates included the following controls: wells containing no cells, wells containing the cells without stimulation, and wells containing the cells with treatments of phorbol 12-myristate 13-acetate (PMA) and Ca ionophore as positive controls. PMA treatment serves as a positive control by mimicking the second messenger, DAG, to activate the T cell receptor pathway and in turn causing T cell activation, with Ca ionophore facilitating entry of calcium ions into cells. The plates were incubated for 36 h at 37 °C before the color reaction. The plates were washed with PBS containing 0.01% Tween 20 six times and then 100 μL detection antibody (clone R64A2, eBioscience, San Diego, CA, USA), diluted in PBS (0.5 μg/mL) was applied to each well. The plates were incubated at 37 °C for an additional 2 h. Plates were again washed with 0.01% Tween 20 in PBS. 100 μL of Streptavidin-AP (1:1000 dilution, Invitrogen) per well was applied and the plates were incubated for 45 min at room temperature. All plates were washed three times again with 0.01% Tween 20 in PBS and PBS alone for another three times. Finally, 100 μL BCIP/NBT (Sigma Aldrich, Munich, Germany) was added to each well for coloration. All plates were washed extensively with tap water and dried before data analysis.

## 5. Conclusions

This study shows that sequential binocular grafts of allogeneic RPCs to the vitreous cavity do not provoke classical immune rejection in the mouse. While it is recognized that these allogeneic histological studies could not be performed with human RPC clinical product, the data appear consistent with the results of a clinical safety study (jCyte, JC-01E, unpublished data) in which patients with retinitis pigmentosa received non-contemporaneous bilateral injections without immunosuppressive treatments. While it is difficult to compare the two studies directly, it is noteworthy that the overall outcomes of both showed similarity in terms of allograft survival.

## Figures and Tables

**Figure 1 ijms-24-06205-f001:**
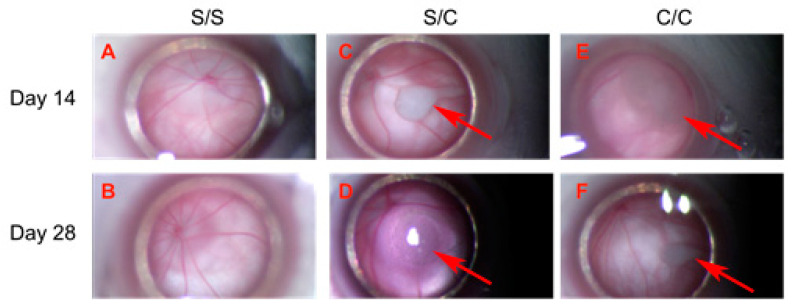
Transplanted gmRPCs visualized in vivo via fundus photography through surgical microscope (all = left eye, OS). (**A**,**B**), sham-treated animals (sham/sham, S/S); (**C**,**D**), control animals with sham-treated right eye followed by gmRPCs-treated left eye (sham/cell, S/C), and animals with both eyes treated sequentially with gmRPCs (cell/cell, C/C) were observed and imaged through the surgical microscope. Upper panel (**A**,**C**,**E**) shows images of OS taken on Day 14 post second injection; lower panel (**B**,**D**,**F**) shows images of OS taken on Day 28 post second injection. Allografts were seen as pale colored cell clumps (arrows) visible in the vitreous of cell-injected left eyes (S/C, C/C) at both time points (Day 14: (**C**,**D**); and Day 28: (**E**,**F**)). Apart from grafts, there was no obvious inflammation or additional abnormality seen in the anterior and posterior segments, as viewed axially in this manner.

**Figure 2 ijms-24-06205-f002:**
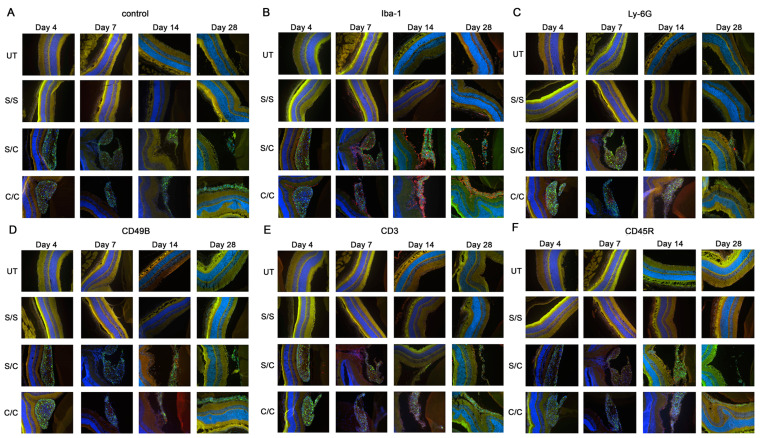
Immuno-labeling of infiltrated immune cells by treatment group. Immunofluorescent images of (**A**) control sections (secondary antibody alone), (**B**) anti-Iba-1 (activated macrophage and microglia marker), (**C**) anti-Ly-6G (neutrophil marker), (**D**) anti-CD49b (natural killer cell marker), (**E**) antibodies anti-CD3 (T cell marker), and (**F**) anti-CD45R (B cell marker) are displayed in the figure. The limited red signals present indicate positive antibody labeling for leucocyte markers, green signals indicate the gmRPC grafts, and blue signals indicate DAPI nuclear labeling. Yellow is nonspecific overlap of signals, most obvious in the photoreceptor outer segments which exhibit autofluorescence. Treatment groups: untreated (UT), both eyes untreated; S/S, both eyes sham treated (sequentially) with vehicle; S/C, right eye injected with vehicle followed by left eye with 50,000 gmRPCs and; C/C, both eyes injected with 50,000 gmRPCs (sequentially) per schedule in Section 4.

**Figure 3 ijms-24-06205-f003:**
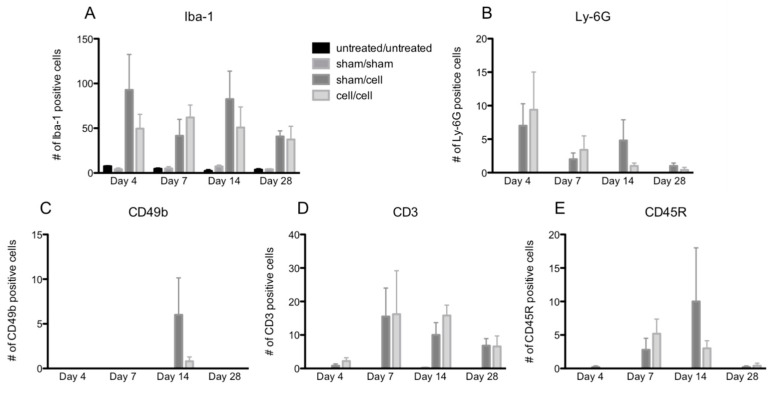
Quantification of immune cell infiltration following gmRPC transplantation. Infiltrated immune cells were visualized as red fluorescent signals within the imaging field following gmRPC injection for each experimental condition, counted using ImageJ, and plotted in bar graphs. Data from Iba-1 (activated macrophage and microglia marker) (**A**), anti-Ly-6G (neutrophils marker) (**B**), anti-CD49b (natural killer cells marker) (**C**), antibodies anti-CD3 (T cells marker) (**D**), and anti-CD45R (B cells marker) (**E**) were displayed in the figure. Two-way ANOVA tests were used to test significance between the sham/cell and cell/cell groups; *p* values were: (**A**) *p* < 0.4492, (**B**) *p* < 0.9376, (**C**) *p* < 0.1190, (**D**) *p* < 0.6411, and (**E**) *p* < 0.7201 (none were significant).

**Figure 4 ijms-24-06205-f004:**
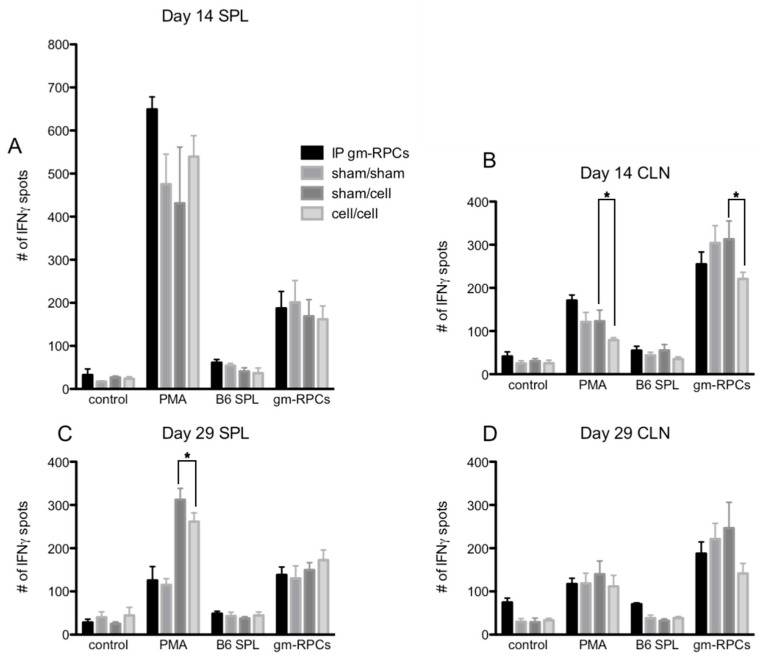
ELISPOT quantification following gmRPC transplantation. ELISPOT assays were setup as the following groups: control, responder cells alone (lymphocytes or splenocytes containing T cells isolated from cervical lymph nodes (CLNs) or spleens (SPL) of the experimental animals; PMA, responder cells treated with phorbol 12-myristate 13-acetate (PMA) and Ca ionophore; B6 SPL, responder cells mixed with splenocytes isolated from C57BL/6 animals; gmRPCs, the same responder cells mixed with gmRPCs. (**A**), The responder cells were the splenocytes from the experimental animals that were sacrificed at day 14 after the second gmRPCs injection. (**B**) The responder cells were the lymphocytes from the CLNs of the experimental animals that were sacrificed at day 14 after the second gmRPCs injection. (**C**) The responder cells were the splenocytes from the experimental animals that were sacrificed at day 29 after the second gmRPCs injection. (**D**) The responder cells were the lymphocytes from the CLNs of the experimental animals that were sacrificed at day 29 after the second gmRPCs injection. Two-way ANOVA tests were used to test significance (*) between the sham/cell and cell/cell groups; *p* values were: (**A**) *p* < 0.5332, (**B**) *p* < 0.0001 (significant), (**C**) *p* < 0.0207 (significant), and (**D**) *p* < 0.3355.

**Table 1 ijms-24-06205-t001:** Experimental groups.

	Intravitreal Injections	
	1st Injection Right Eye (OD)	OS Injected 2 Weeks after OD Injection
**Group 1 (*n* = 3 per time point)**	untreated	untreated
**Group 2 (*n* = 5 per time point)**	sham	sham
**Group 3 (*n* = 5 per time point)**	sham	50,000 GFP mouse-RPC
**Group 4 (*n* = 5 per time point)**	50,000 GFP mouse-RPC	50,000 GFP mouse-RPC
	**Intraperitoneal Injections**	
**Positive Control 1 (*n* = 3)**	untreated	10^6^ RPC intraperitoneal
**Positive Control 2 (*n* = 3)**	10^6^ RPC intraperitoneal	10^6^ RPC intraperitoneal

**Table 2 ijms-24-06205-t002:** Evaluation time points.

Timepoint Evaluation	Day 4	Day 7	Day 14	Day 28
Histopath/IF	Groups 1–4	Groups 1–4	Groups 1–4	Groups 1–4
ELISPOT	none	none	Groups 2–4 Positive Controls	Groups 2–4 Positive Controls

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
