# Peer review of "Immune Responses to Sequential Binocular Transplantation of Allogeneic Retinal Progenitor Cells to the Vitreous Cavity in Mice"

_ijms, 2023, doi:10.3390/ijms24076205_

Round 1

Reviewer 1 Report (Previous Reviewer 2)

While the authors have done a spectacular job of addressing the concerns the response to concern #1 was misinterpreted by the authors. It was not whether the injections induced a regulatory immune response, it was that the injection will induce a regulatory response. It is the injection of foreign antigen within the eye induces systemic tolerance to the antigen. It is that response that influences all aspects of the results. This is not a species-dependent mechanism. It is an evolutionarily conserved mechanism in all mammals with examples in humans too. 

Author Response

We thank the reviewer for their appreciation of our responsiveness and for their clarification of prior comments.

Reviewer 2 Report (New Reviewer)

Abstract

Was it only investigated the immuno response of graft injection or as well the effect of allogeneic human retinal progenitor cells (hRPCs)?

Results

2.1 How potential of inflammation was measured? Is there any reference with inflammation to show the exact mechanism to suggest no inflammation appear in the present study.

- indicate e.g. as arrow in pictures of fig 1 ”animals with both eyes treated sequentially with gmRPCs.. Pale colored cell clumps (allografts) were visible”

- Fig 1. Indicate anterior and posterior parts eye.

- Figure 2. Pictures could be larger and indicate if there is red marker as visible or not at all.

- Indicate statistics in Fig 3.

- Fig 3, Why compared only groups sham/cell and cell/cell groups? and what timepoint? Why does not compared sham/sham to both of those groups?

-open all abbreviations when mentioned first time

- line 181, what means that alternative stimulator cells? Need to open because response is so different that it is confusing when continue to read and until line 183 where response is explained as high background. Is the treatment same than others or different? now it is confusing if this is repetition and results are different? Modify to avoid misunderstanding. Results need to be clear and if opposite the situation is not stable or repeatable. Or is that only other cell line, plenocytes? Open little bit more to be more clearer for reader.

- Figure 4 is complex. There is many abbreviations not yet so familiar from the beginning of the manuscript, as well different treatment groups. Is it possible to devide figure example into two different figures and open little bit more clearly all abbreviations in the pictures into the figure legend. Example groups under x-line are not yet clear for reader as example treatment groups are.

- What is PMA? Is it some kind of positive control?

- Conclude at the beginning of the result part at least in 2.3 the cell that were investigated. Now it is little bit too much for reader to figure out all and it could be better to conclude shortly all investigated cells especially when material and methods part is placed into the end after manuscript.

- There is in figure 4 legend statistical differences in PMA group. That should open little bit more. As well in figure there is a lot of information and part of it is opened only little bit in the text. It should be better to open little bit more results of different groups in the text even there is not significant responses at least that groups (control, PMA, B6 SPL, gm-RPCs) comes more familiar to the reader.

- Figure 3. As well sham/cell and cell/cell groups should statistically at least compared to sham/sham group.

- If injections are in reality to be gived more than twice it could be nice to at least continue treatments little bit more than just two times. How about immune reaction in the eye? If treatments are related to retinitis pigmentosa I assumed that injection were intravitreal injectios? Why immune responses in the eye were not investigated? Probably first immune reaction start in the locally in place where compounds are injected. Could immune reaction be local and not systemic?

 Discussion

-Discussion start nicely and that kind of reason and background for the study could come already from the beginning of the introduction.

Material and Methods

- line 298, change RPM to g-value.

- complement the reagent informations

- line 423, explain little bit background of positive controls.

Author Response

Abstract

Was it only investigated the immuno response of graft injection or as well the effect of allogeneic human retinal progenitor cells (hRPCs)?

Correct. Only murine RPCs were investigated here. This is now clarified at the end of the Introduction. Other studies, including human RPCs, are referenced for context.

Results

2.1 How potential of inflammation was measured? Is there any reference with inflammation to show the exact mechanism to suggest no inflammation appear in the present study.

A basic descriptive judgement made by clinical inspection via surgical-type microscope. More detailed histological evidence of cellular response is of course also of interest, as addressed via IHC on tissue sections as shown.

- indicate e.g. as arrow in pictures of fig 1 ”animals with both eyes treated sequentially with gmRPCs.. Pale colored cell clumps (allografts) were visible”

Done

- Fig 1. Indicate anterior and posterior parts eye.

The anterior part of eye as seen in these fundus photos is transparent so difficult to label, but we have added additional clarifying language to the Figure Legend.

- Figure 2. Pictures could be larger and indicate if there is red marker as visible or not at all.

The was some limited red labelling, which has now been addressed in the Figure Legend and is detailed in the Results 2.2 section. We will work with the Journal to optimize Figure size.

- Indicate statistics in Fig 3.

For Figure 3, it was 2-way ANOVA, none were significant, as stated in Fig. 3 Legend.

- Fig 3, Why compared only groups sham/cell and cell/cell groups? and what timepoint? Why does not compared sham/sham to both of those groups?

The immune response following repeat dosing is the central question here and has been evaluated across all experimental timepoints using ANOVA. A mild response to single dosing with allogeneic RPCs is beyond dispute, and evident from the Figures, so statistical analysis would be largely redundant.

-open all abbreviations when mentioned first time.        

Done

- line 181, what means that alternative stimulator cells? Need to open because response is so different that it is confusing when continue to read and until line 183 where response is explained as high background. Is the treatment same than others or different? now it is confusing if this is repetition and results are different? Modify to avoid misunderstanding. Results need to be clear and if opposite the situation is not stable or repeatable. Or is that only other cell line, plenocytes? Open little bit more to be more clearer for reader.

The gmRPCs are the alternative stimulator cells, being the “alternative” to control splenocytes of the same genetic background (C57BL/6). The key finding is not the overall background elevation of the gmRPC treatment groups, but rather the lack of difference in response between individual gmRPC treatment groups, as mounted by responder cells of the host immune system (and measured as ELISPOTs). A positive response would be seen as a markedly higher bar for cell/cell as compared to sham/cell in the gmRPC category. But that is not what we see. If anything, retreatment with gmRPC may in some cases have a negative impact on host lymphocyte responsiveness (e.g., 4B), but over-all the conclusion is that retreatment with gmRPCs does not provoke an amplified immune cell activation.

We have worked on the text to help communicate the message. Also see our related responses below.

- Figure 4 is complex. There is many abbreviations not yet so familiar from the beginning of the manuscript, as well different treatment groups. Is it possible to devide figure example into two different figures and open little bit more clearly all abbreviations in the pictures into the figure legend. Example groups under x-line are not yet clear for reader as example treatment groups are.

Yes, Fig. 4 is somewhat complex, and one could say unavoidably so, even though it is arguably the least essential to the overall message (see above). In a different setting like a review paper it might be nice to provide background information in the form of schematic illustrations. Here we have defined the potentially unfamiliar abbreviations CLN, SPL, and PMA in Figure 4 legend, which is getting long at this point. Also, the term gmRPC has now been defined multiple times earlier in the paper.

We have now revised and augmented the text of Results 2.3 and the Methods. These sections will likely be helpful for readers less familiar with immunological techniques, so that they get the most out of Figure 4.

- What is PMA? Is it some kind of positive control?

Yes. PMA = phorbol 12-myristate 13-acetate, a treatment condition used as a positive control mimicking the second messenger, DAG, to activate the T cell receptor pathway and in turn causing T cell activation. PMA is defined in Figure 4 legend, and described (as above) in Results section 2.3.

- Conclude at the beginning of the result part at least in 2.3 the cell that were investigated. Now it is little bit too much for reader to figure out all and it could be better to conclude shortly all investigated cells especially when material and methods part is placed into the end after manuscript.

The cell under investigation is always the donor gmRPCs, but there are also host immune cells involved in the assays that comprise the readout (or serve as controls). Results 2.3 has been reworked to better highlight the donor cell results so they don’t get lost in the “forest” of controls.

- There is in figure 4 legend statistical differences in PMA group. That should open little bit more. As well in figure there is a lot of information and part of it is opened only little bit in the text. It should be better to open little bit more results of different groups in the text even there is not significant responses at least that groups (control, PMA, B6 SPL, gm-RPCs) comes more familiar to the reader.

Results 2.3 has been augmented as described elsewhere. The PMA results referred to are not of particular interest to the thrust of this report and expounded on them would generate additional distraction from the message.

- Figure 3. As well sham/cell and cell/cell groups should statistically at least compared to sham/sham group.

See above.

- If injections are in reality to be gived more than twice it could be nice to at least continue treatments little bit more than just two times. How about immune reaction in the eye? If treatments are related to retinitis pigmentosa I assumed that injection were intravitreal injectios? Why immune responses in the eye were not investigated? Probably first immune reaction start in the locally in place where compounds are injected. Could immune reaction be local and not systemic?

Yes, the clinical plan calls for multiple injections, although no patient has received more than 2 injections to date. Thank you for the suggestion. We do consider investigating more than 2 injections in mice.

Yes, in RP patients (data not shown in the present report), the cells were injected into the vitreous cavity. The patients underwent careful examination as part of the FDA-registered clinical trials. Yes, the studies were much more focused on local immune response than systemic, for multiple reasons, including the fact that the graft site can be serially inspected via in vivo biomicroscopy. Immune rejection was not observed, and that is something we reference here but do not include in this report. However, it is the case that the novel clinical trial reports are being written up separately.

The investigation of intraocular immune response reported here includes IHC data as well as in vitro results and therefore sets the stage for the upcoming clinical reports of allografts in humans. 

Discussion

-Discussion start nicely and that kind of reason and background for the study could come already from the beginning of the introduction.

These points from Discussion have been strengthened in the Intro

Material and Methods

- line 298, change RPM to g-value.     Done
- complement the reagent informations     Done
- line 423, explain little bit background of positive controls.   Done                    

This manuscript is a resubmission of an earlier submission. The following is a list of the peer review reports and author responses from that submission.

Round 1

Reviewer 1 Report

Reviewer

Comments to the Author
The paper by Chen et al studied repeated intravitreal retinal progenitor cell (RPC) transplantation into mice, in order to test any induce immune response. In their study showed no ocular abnormalities, but observed mild immune response upon repeated cell transplant injection. This was an interesting experimental design and they asked relevant scientific research questions for the field of cell transplantation. Herein I have some additional comments to further consider.

11. The introduction contains relevant literature and has good summary for their hypothesis to test.

22. How is the 2-week delayed second injection of RPCs equivalent with the human sequential injections time? Would more time left between the different injections results in no immune response at all? Are there plans to test this hypothesis? This could be added to their discussion.

33. The authors observed dominantly Iba-1 positive cell appearance upon RPC injection. Is this response specific to mice or there are human studies showing also macrophage response upon cell transplantation? This could be added to their discussion.

  4. Significance should be indicated on the figures with at least *. For the reader, it is misleading, when looking at the graph and significance values can only be found ini the figure legends.

55. In their discussion, they claim, that there was no decrease in graft viability upon repeat dosage of RPCs. What is the ‘viability’ statement based upon? Do they base this on Figure 1? If they used GFP-RPCs in their experiment, did not they take fluorescent imaging of the mice eyes? Please explain this, or add additional figures to prove no decrease in cell viability.

66. Their discussion does not contain any references added. Their results has to be evaluated in the point of view of existing literature. The discussion with relevant references and comparisons to literature must be improved before further acceptance of the manuscript.

Reviewer 2 Report

This manuscript demonstrated that repeated transplants of allogenic retinal progenitor cells into the vitreous cavity survived and that there is a low or no immune response to the transplanted cells. The authors have concluded that the immune privileged nature of the eye protects the allografts and have done an excellent job of histological and immune analysis of the transplant. However, the methods used to show sequential transplantation are no different from the methods used to show an Anterior Chamber Associated Immune deviation (ACAID)-like response demonstrated by Jiang and Streilein in the 1990s and later reports from Streilein on retinal mediated immune regulation.

1. The experiments need to show that you have induced regulatory immunity.

2. The experiments should be repeated in mice with retinal degeneration. In these mice, the immune privilege of the ocular microenvironment is compromised, unlike using healthy mice with immune privilege intact.

Reviewer 3 Report

This manuscript is seriously lacking in quality, relevance, and scientific rigor on all areas.  I cite several here:

1)    No histological examination of grafted cells, even though GFP+ cells were used as donor material, this is unacceptable, and totally invalidates any of the so-called conclusions the authors make.

2)    The most self-serving citations this reviewer has ever seen, unacceptable.  This of course has the corollary of a complete lack of review of the relevant experience of the field which is extensive and completely ignored.

3)    The experimental design lacks rationale for any conclusions to be made re: the human trials the authors are involved in, yet conclusions are made relative to this work, also unacceptable.  

Reviewer 4 Report

1. What cells were used for analysis? vitreous cavity? or blood?
If cells in vitreou cavity were used, mixture of gmRPC and immune cells were used for analysis?

2. wasn't there tumor formation?

3. Figure 3. Looks the significance. Number in each group was too small??

Figure 4. PMA samples did not show any differecnce from cotrol?

4. Single cell sequencing for immune cells is needed.